# BRIDGE FRAME AND EVENT: COMMON SPATIOTEMPORAL FUSION FOR HIGH-DYNAMIC OPTICAL FLOW

## ABSTRACT

High-dynamic scene optical flow is a challenging task, which suffers large displacement. Limited by frame imaging, large displacement causes potential spatial blurry textures due to long exposure and temporal discontinuous motion due to low frame rate, thus deteriorating the spatiotemporal feature of optical flow. Typically, existing methods mainly introduce event camera with high temporal resolution to directly fuse the spatiotemporal features between the two modalities. However, this direct fusion is ineffective, since there exists a large gap due to the heterogeneous data representation between frame and event modalities. To address this issue, we explore a common-latent space as an intermediate bridge to mitigate the modality gap. In this work, we propose a novel common spatiotemporal fusion between frame and event modalities for high-dynamic scene optical flow, including visual boundary localization and motion correlation fusion. Specifically, in visual boundary localization, we figure out that frame and event can be derived into the spatiotemporal gradient maps with the same data representation, where the similarity distribution between the two modalities is consistent with the extracted boundary distribution. This motivates us to design the common spatiotemporal gradient to constrain the localization of the reference boundary as a template. In motion correlation fusion, we discover that the frame-based motion possesses spatially dense but temporally discontinuous correlation, while the event-based motion has spatially sparse but temporally continuous correlation. This inspires us to take the reference boundary template to guide the fusion of the complementary motion knowledge between the two modalities. Moreover, common spatiotemporal fusion can not only relieve the cross-modal feature discrepancy, but also make the fusion process interpretable to achieve dense and continuous optical flow. Extensive experiments have been performed to verify the superiority of the proposed method.

## 1 INTRODUCTION

Optical flow is a task of modeling the pixel-level temporal correspondence from the spatial features between adjacent frames. Optical flow has made great progress in conventional dynamic scenes with ideal imaging, but paid little attention to the challenging high-dynamic scenes, *i.e.*, fast motion. The main reason is that fast motion in the world space brings in the degradation of large displacement to the frame-based imaging space, thus breaking the basic spatiotemporal assumption of optical flow in Fig. 1 (a). On the one hand, large displacement leads to the spatial blur of visual textures due to long exposure, violating the photometric constancy assumption. On the other hand, large displacement causes the temporal discontinuity of motion trajectory due to low frame rate, breaking the gradient continuity assumption. The two degradations further deteriorate the spatiotemporal features of optical flow in the feature space. In this work, we focus on modeling the spatiotemporal characteristic of high-dynamic scene motion, thus achieving spatial-dense and temporal-continuous optical flow.

From the perspective of imaging mechanism, event camera (Gallego et al., 2020) is a neuromorphic vision sensor with asynchronous trigger, which has the advantage of high temporal resolution to make up for the limitation of frame camera. Regarding the issue of high-dynamic scene optical flow, several methods (Wan et al., 2022; Gehrig et al., 2024) introduce event camera, and formulate this problem as a task of multimodal fusion, which directly fuses the spatiotemporal features between frame and event modalities to achieve dense and continuous optical flow in Fig. 1 (b). For example, Wan et al. (2022) encoded the frame images and event stream into the spatial features and the temporal features,

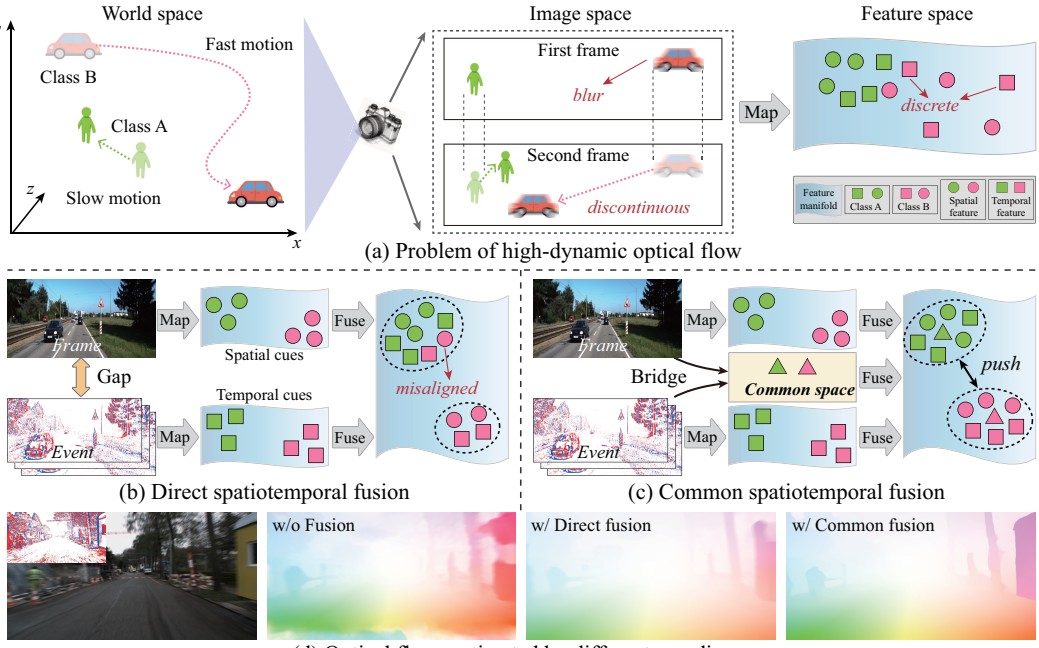

(a) Problem of high-dynamic optical flow

(b) Direct spatiotemporal fusion

(c) Common spatiotemporal fusion

(d) Optical flows estimated by different paradigms

Figure 1: Illustration of problem and idea. High-dynamic target in world space cause degraded motion with spatial blur and temporal discontinuity in image space, resulting in the discretization of spatiotemporal features in feature space. Therefore, high-dynamic optical flow is formulated as a problem of reinforcing the spatiotemporal features. Direct spatiotemporal fusion introduces event camera to compensate for frame camera, and directly fuses their spatiotemporal features. However, this direct fusion suffers the feature misalignment, since there exists a large gap due to the heterogeneous data representation between frame and event modalities. In this work, we explore the common space to bridge the gap, thus guiding the frame-event spatiotemporal feature fusion.

and then concatenated them to iteratively update optical flow. However, this direct fusion solution is ineffective, since there exists a large gap due to the heterogeneous data representation and feature distribution between frame and event modalities. Therefore, *building a common-latent space to bridge the modality gap is crucial for the spatiotemporal fusion of optical flow.*

To address this issue, we explore a common feature space as an intermediate bridge to mitigate the feature discrepancy between frame and event modalities in Fig. 1 (c), thus guiding their spatiotemporal fusion. As for the common space, we theoretically derive that the frame and event modalities can be transformed into the spatiotemporal gradient map with the same data representation via different formulas. We also figure out that the similarity distribution of the spatiotemporal gradient maps between the two modalities is consistent with the extracted boundary distribution, which motivates us to take the spatiotemporal gradient as the common-latent space to locate the reference points from the boundary features. As for the spatiotemporal fusion, we encode the frame and event data into the motion feature space, where we discover that the frame-based motion feature possesses spatially dense but temporally discontinuous correlation, while the event-based motion feature has spatially sparse but temporally continuous correlation. This inspires us to choose the reference boundary points as the template to guide the complementary spatiotemporal fusion of motion knowledge between the two modalities. Therefore, the common spatiotemporal gradient space can serve as an intermediate bridge to advance the cross-modal spatiotemporal correlation fusion.

In this work, we propose a novel commonality-guided spatiotemporal fusion (ComST-Flow) framework between frame and event modalities for high-dynamic scene optical flow in Fig. 2, including visual boundary localization and motion correlation fusion. Specifically, we first collect the pixel-aligned frame images and event stream with coaxial optical device. During visual boundary localization stage, we transform the frame images and event stream into the common spatiotemporal gradient space, where we calculate the similarity distribution between the two modalities. We then utilize feature pyramid (Lin et al., 2017) to extract the boundary features of the frame images and

event stream, and take the similarity distribution of the common spatiotemporal gradient to locate the reference points of the boundary features. During motion correlation fusion stage, we map the visual features of frame and event modalities into the motion correlation space. We employ the located reference boundary template to guide the spatial feature clustering within the frame-based correlation for the density of motion, and the temporal feature tracking within the event-based correlation for the continuity of motion. We finally introduce the cross-modal transformer architecture to fuse the spatiotemporal correlation features between frame and event modalities. Under this unified framework, the proposed common spatiotemporal fusion can close the modality gap and explicitly fuse the spatiotemporal motion knowledge between frame and event modalities, thus achieving dense and continuous optical flow. Overall, our main contributions are summarized:

- We propose a novel commonality-guided spatiotemporal fusion framework between frame and event modalities for high-dynamic scene optical flow, which can close the modality gap and make the cross-modal fusion process interpretable to achieve dense and continuous optical flow.

- We construct a common spatiotemporal gradient space as an intermediate bridge to mitigate the heterogeneous data representation and feature distribution between frame and event modalities, thus improving the spatiotemporal fusion between the two modalities.

- We reveal that the frame-based motion feature possesses spatially dense but temporally discontinuous correlation, while the event-based motion feature has spatially sparse but temporally continuous correlation, motivating us to fuse the frame-event complementary spatiotemporal correlation.

- We propose a pixel-aligned frame-event dataset with optical coaxial device, and conduct extensive experiments to demonstrate that the proposed fusion method achieves state-of-the-art performance for high-dynamic scene optical flow.

## 2 RELATED WORK

**Optical Flow.** In recent years, learning-based methods have been proposed to model the motion features of optical flow, including supervised and unsupervised methods. Supervised methods (Dosovitskiy et al., 2015; Luo et al., 2022; Sun et al., 2018; Ilg et al., 2017; Hur & Roth, 2019; Teed & Deng, 2020; Lu et al., 2023; Jiang et al., 2021) usually design different deep networks to learn motion field using optical flow labels. Furthermore, to relieve the dependency on motion labels, unsupervised methods (Jonschkowski et al., 2020; Liu et al., 2019a;b; Meister et al., 2018; Ren et al., 2017; Yu et al., 2016) mainly employ the basic spatiotemporal assumption (*i.e.*, photometric constancy and gradient continuity) of optical flow as the prior to design the training losses for motion feature representation. However, despite these methods could satisfy conventional dynamic scenes with ideal imaging, they are still limited by the degradation caused by large displacement in frame imaging under high-dynamic conditions. The main reason is that large displacement leads to the spatial blur of visual textures and the temporal discontinuity of motion trajectory, thus breaking the basic spatiotemporal assumption of optical flow. In this work, we aim to improve the spatiotemporal feature representation, thus achieving spatial-dense and temporal-continuous optical flow.

**High-Dynamic Scene Optical Flow.** In frame imaging, large displacement may cause spatial blur of visual textures and temporal discontinuity of motion trajectory in high-dynamic scenes. An intuitive solution is image preprocessing, such as image deblur (Ruan et al., 2022; Kupyn et al., 2019) and frame interpolation (Plack et al., 2023; Zhang et al., 2023). Image deblur does indeed improve the spatial visualization quality of frame imaging, but cannot guarantee the valid motion feature matching in the temporal dimension. Frame interpolation is indeed beneficial for estimating temporally continuous optical flow, but it is difficult to model the temporally non-linear motion pattern of dynamic targets from the frames with low frame rate. Another novel solution is to introduce event camera with high temporal resolution to assist frame camera in optical flow, including unimodal and multimodal methods. Unimodal methods (Zhu et al., 2018; Paredes-Vallés & de Croon, 2021; Gehrig et al., 2021b) input event stream to learn the temporal continuous optical flow, while is limited by the spatial sparsity of event data. Multimodal methods (Wan et al., 2022; Gehrig et al., 2024) formulate this problem as a fusion task, which directly fuse the spatiotemporal motion features between frame and event modalities for dense and continuous optical flow. However, these direct fusion methods neglect the large gap due to the heterogeneous data representation between frame and event modalities. To address above issue, we explore an intermediate bridge to close the modality gap, thus improving the cross-modal spatiotemporal fusion of motion features.

Figure 2: The architecture of the ComST-Flow contains visual boundary localization and motion correlation fusion. In visual boundary localization, we transform frame images and event stream into common spatiotemporal gradient space. We further constrain the gradient similarity and the extracted boundary similarity between the two modalities, locating the reference boundary points as the template. In motion correlation fusion, we introduce cross-modal transformer to fuse the spatially dense correlation from frame modality and the temporally continuous correlation from event modality under the guidance of the boundary template, thus achieving dense and continuous optical flow.

**Multimodal Fusion.** Multimodal fusion aims to exploit the inter-modal complementary knowledge for the target vision task. To our knowledge, most existing multimodal fusion methods (Hori et al., 2017; Hu et al., 2017; Wei et al., 2020; Sun et al., 2022; Cai et al., 2023) construct the feature-level attention to fuse the cross-modal complementary knowledge. For example, Sun et al. (2022) built a cross-modal attention module to fuse the relevant features between frame and event modalities for motion deblurring using several residual convolution blocks. However, these multimodal fusion methods face two problems. First, there exists a large gap between different modalities, exacerbating the discrepancy of the cross-modal feature distribution, thus limiting the fusion performance. Second, the complementary features lack the physical meanings, and the entire fusion process is too implicit, causing uncontrollable fusion results. In this work, we design a common-latent space with physical meanings to bridge the frame-event gap, as a guidance to fuse the complementary spatiotemporal correlation for motion representation, thus controllably achieving dense and continuous optical flow.

# 3 COMMON SPATIOTEMPORAL FUSION

## 3.1 OVERALL FRAMEWORK

High-dynamic scene optical flow suffers the degradations caused by large displacement of fast motion in the spatial and temporal dimensions. The key of solving this issue is how to improve the spatiotemporal feature representation of the motion. To this end, we rethink this problem from the perspective of multimodal fusion, and introduce event camera to assist frame camera in the spatiotemporal fusion of motion features, thus better representing optical flow. Considering that there exists a large gap due to the heterogeneous data representation between frame and event modalities, we further explore a common-latent space to bridge the modality gap, thus alleviating the feature discrepancy between the two modalities. In this work, we propose a novel commonality-guided spatiotemporal fusion framework between frame and event modalities for high-dynamic scene optical flow. As shown in Fig. 2, the whole framework looks complex but is simplified into two sub-modules, *i.e.*, visual boundary localization and motion correlation fusion. In visual boundary localization module, we first build the common spatiotemporal gradient space, where we can represent both frame and event modalities using the same data form. Next, we utilize the common space to constrain the reference point localization of the boundary features fused from the two modalities. In motion correlation fusion module, we take the located reference points of the boundary as the template to guide the feature clustering within the frame-based correlation for the density of motion appearance in the spatial dimension, and guide the feature tracking within the event-based correlation for the continuity of motion trajectory in the temporal dimension. Finally, we build the cross-modal transformer architecture to fuse the spatiotemporal correlation features. Under this unified framework, visual boundary localization constructs common spatiotemporal gradient space to bridge frame and event, while motion correlation fusion fuses the complementary spatiotemporal motion knowledge under the guidance of the common space, thus achieving dense and continuous optical flow.

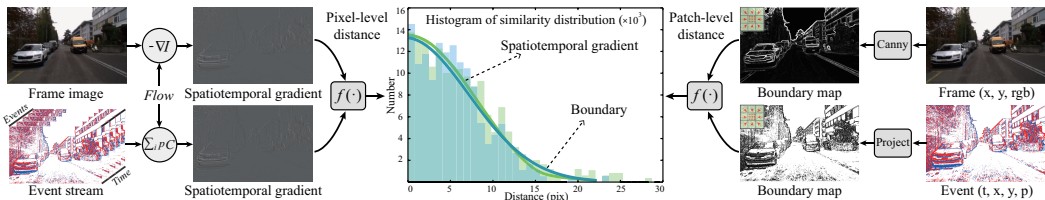

Figure 3: Similarities of the spatiotemporal gradient and boundary between frame and event modalities. We use Euclidean distance to calculate the frame-event spatiotemporal gradient similarity and the boundary similarity. The distributions of the two similarities are consistent, which motivates us to take spatiotemporal gradient as the common space to constrain the localization of boundary points.

## 3.2 Visual Boundary Localization

Estimating optical flow in high-dynamic scenes is challenging, since large displacement of fast motion causes spatiotemporal degradations due to the limitation of frame imaging. Therefore, we introduce event camera with high temporal resolution to assist frame camera in fusing the complementary knowledge for dense and continuous optical flow. However, there are two difficulties: pixel-level alignment and large modality gap between frame and event cameras.

**Pixel-Aligned Frame and Event.** The pixel-level alignment between frame and event cameras heavily relies on spatial calibration (*e.g.*, hardware and software) and time synchronization (see supplementary for details). Hardware calibration sets up a physically coaxial optical device with a beam splitter for frame and event cameras, which allows the same light to pass through the same lens and enter different cameras. Software calibration further performs a standard stereo rectification between frame data and event data, and then refines the slight calibration errors via pixel offset (Tulyakov et al., 2022). Moreover, time synchronization is achieved by external trigger from microcontroller. In this way, we can obtain the spatiotemporal pixel-aligned frames and event stream.

**Common Spatiotemporal Gradient Space.** To explore the common space for the frame-event gap, we extend the optical flow basic model (Paredes-Vallés & de Croon, 2021) via Taylor expansion:

$$I(x + dx, t + dt) = I(x, t) + (\frac{\partial I}{\partial x}dx + \frac{\partial I}{\partial t}dt) + O(dx, dt), \tag{1}$$

where $dx$ denotes pixel displacement, and $dt$ denotes duration. Then, we remove the high-order error term $O(dx, dt)$ to approximate Eq. 1 as follows:

$$I_t = -\nabla I \cdot U, \tag{2}$$

where $U$ denotes the optical flow estimated from frame images and event data via the pre-trained flow model (Huang et al., 2022). $\nabla I$ is the spatial gradient field of the frame. $I_t$ denotes brightness change along the time dimension, which can be approximated as the accumulated events warped by the optical flow in a certain time window as follows:

$$I_t = U \cdot \sum_{e_i \in dt} p_i C, \tag{3}$$

where $e_i$ is the event timestamp, $p_i$ is the event polarity, and $C$ denotes the event trigger threshold. Eq. 2 and Eq. 3 indicate that frame and event modalities can be ideally transformed into the consistent spatiotemporal gradient with the same data representation through various formulas, which can serve as a common-latent space to bridge the gap between frame and event modalities.

**Boundary Template Localization.** Optical flow learns the correspondence between spatial visual features at different timestamps. This can be approximated as temporal template matching. As we know, frame reflects global appearance while event senses local boundary. We use Euclidean distance to measure the similarity between the two modalities within common spatiotemporal gradient space:

$$C = f(U \cdot \sum_{e_i \in dt} p_i C, -\nabla I \cdot U), \tag{4}$$

where $f(\cdot)$ is the similarity metric function with Euclidean distance. And then, we utilize the canny operator to extract the 2D boundary map from the frame image, and temporally project the event data into the 2D boundary map. We take sliding window to divide the frame-based and the event-based 2D boundary maps into many corresponding patches, and we also use Euclidean distance to calculate the similarities among all the boundary patches. Furthermore, we obtain the similarity distributions of the common spatiotemporal gradient and the extracted boundary via histograms in Fig. 3, which shows that the two similarity distributions are consistent. This motivates us to take common spatiotemporal

Figure 4: Correlation distribution of frame and event modality. We transform frames and events to correlation volume via warping, and cluster the sampled correlation features. The frame-based correlation features are x, y-axis spatially dense, while the event-based correlation features are t-axis temporally continuous. This inspires us to fuse the complementary spatiotemporal correlation between the two modalities for dense and continuous optical flow.

gradient space to constrain the extraction of visual boundary features. Specifically, we first transform frame images and event stream into the common spatiotemporal gradient space, and calculate the similarity distribution $p_0$ between the two spatiotemporal gradient maps. Next, we encode adjacent frames and $T$ event slices into visual feature space, where we introduce a shared feature pyramid network to extract the corresponding boundary features. We then utilize convolutional kernel to generate the similarity distribution $p$ between the boundary features of the two modalities, and make the boundary distribution close to the gradient similarity $p_0$ via K-L divergence as follows:

$$\mathcal{L}_{kl} = \sum p \, log \frac{p}{p_0}. \tag{5}$$

Moreover, we further introduce softmax function to generate the cross-modal boundary map $P$ with probability via the cross-entropy loss as follows:

$$\mathcal{L}_{entropy} = -\mathbb{E} \sum\nolimits_{i,j \in P} \sum\nolimits_{k=1}^{K} \mathbb{I}_{[k=\mathbf{y}]} log(P(i,j)), \tag{6}$$

where the probability value $log(P(i,j))$ can reflect the degradation degree of the frame. $K$ denotes the boundary class number according to the probability range, where $0$ corresponds to the normal boundary. $\mathbf{y}$ is the pre-classified boundary class label. Finally, we filter the boundary points with low probability, namely severely degraded points, and the filtered boundary points are used as the reference template to guide the sequent frame-event spatiotemporal fusion.

### 3.3 MOTION CORRELATION FUSION

Although visual boundary localization can bridge the large modality gap and provide the visual boundary points as the reference template, selecting which feature to use for fusion and how to utilize the template to guide this fusion for optical flow is a challenge.

**Complementary Frame-Event Spatiotemporal Correlation.** As for the choice of the features, the motion features that optical flow relies on mainly include two categories: correlation feature and decoded feature. Correlation feature is derived from visual features via warping and pixel-wise multiplication, namely correlation volume, which represents the temporal correspondence of visual features. Decoded feature is updated by recurrent network (*e.g.*, GRU (Cho et al., 2014)). Therefore, correlation feature has a physical meaning, which could serve as the key motion feature to fuse. As for the frame-event complementary nature, frame reflects dense appearance while event senses sparse boundary in the spatial dimension, and frame captures discontinuous displacement while event triggers continuous displacement in the temporal dimension. This naturally makes us wonder whether these two modalities also have similar complementary knowledge at the level of motion features. To demonstrate this insight, we first extract the boundary corner as the template, and then transform frames and event stream to correlation volume space via warping, where we sample the correlation features to match this template using K-Means clustering (Likas et al., 2003). As shown in Fig. 4, the frame-based corelation features are x, y-axis spatially dense but t-axis temporally discontinuous, while the event-based correlation features are x, y-axis spatially sparse but t-axis temporally continuous. Therefore, we choose correlation feature as the motion feature, and fuse the frame-based spatially dense correlation and the event-based temporally continuous correlation.

Table 1: Quantitative results on synthetic Event-KITTI and real DSEC datasets.

| Method | ARFlow | Selflow | | RAFT | GMA | | E-RAFT | BFlow | ComST-Flow |
|---|---|---|---|---|---|---|---|---|---|
| | | − | w/ Deblur | | − | w/ Deblur | | | |
| Input | Frame | Frame | Frame | Frame | Frame | Frame | Event | Frame-Event | **Frame-Event** |
| Event- EPE | 33.52 | 8.03 | 6.89 | 0.87 | 0.79 | 0.74 | 2.48 | 0.76 | **0.72** |
| KITTI F1-all | 83.45% | 26.13% | 23.15% | 5.34% | 4.85% | 4.36% | 9.37% | 4.35% | **3.83%** |
| Slow- EPE | 14.81 | 15.46 | 14.24 | 0.97 | 0.93 | 0.92 | 0.95 | 0.87 | **0.47** |
| DSEC F1-all | 70.54% | 72.27% | 71.24% | 4.15% | 3.78% | 3.53% | 3.68% | 2.98% | **1.17%** |
| Fast- EPE | 15.60 | 16.16 | 15.44 | 1.35 | 1.24 | 1.22 | 0.95 | 0.87 | **0.58** |
| DSEC F1-all | 72.47% | 78.07% | 71.57% | 6.26% | 5.12% | 4.96% | 3.65% | 2.89% | **1.96%** |

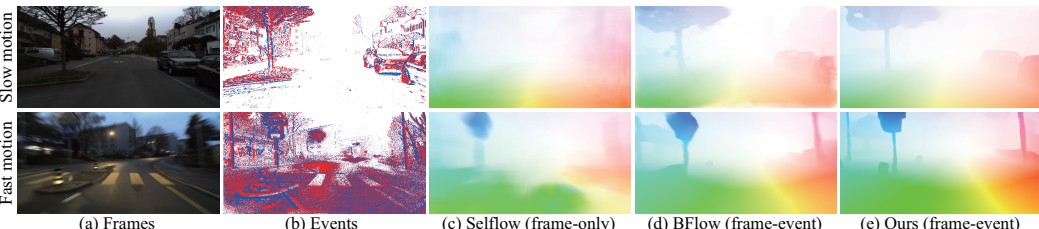

Figure 5: Visual comparison of optical flows on real DSEC dataset with slow and fast motion.

**Correlation Spatiotemporal Fusion.** The scene contains different dynamic targets with various motion patterns, thus simple global fusion may lead to invalid correlation representation regardless of motion patterns. This requires that we should focus more on the spatiotemporal correlation fusion of the dynamic target with the same motion pattern, not different dynamic targets. Therefore, we utilize the reference template to guide the spatial matching of the frame-based correlation and the temporal matching of the event-based correlation, respectively, where these two correlation features belong to the same dynamic target. In the spatial matching, we take the point coordinates of the reference template as the center, and introduce K-Means algorithm to cluster the correlation features near the center in the spatial dimension, which is matched with the correlation feature located by the reference template. Spatial correlation matching assigns the correlation features with region attribute, like neighborhood in the image, thus promoting the spatially dense motion appearance. In the temporal matching, we introduce the extended Kalman filter (EKF) to track the correlation features located by the reference template. Given the reference template, we initialize the motion state vector of the reference template consisting of coordinate, displacement and correlation, and utilize the standard EKF algorithm to update the motion state vector at different timestamps, thus forming the temporally continuous motion trajectory. We then introduce transformer architecture (Vaswani et al., 2017) to model the spatiotemporal fusion process between frame and event modalities, which outputs dense and continuous correlation features. For spatial density, we constrain the correlation spatial error between the estimated correlation $cv$ and the dense correlation $cv^{spa}$ from the spatial matching:

$$\mathcal{L}_{corr}^{spaErr} = \frac{1}{N} \sum_{n=0}^{N} ||cv_{t=0,n} - cv_n^{spa}||_1, \tag{7}$$

where $N$ is reference boundary point number. For temporal continuity, we constrain the temporal error between the correlation $cv$ and the continuous correlation $cv^{temp}$ from the temporal matching:

$$\mathcal{L}_{corr}^{tempErr} = \frac{1}{T} \sum_{t=0}^{T} ||cv_t - cv_t^{temp}||_1. \tag{8}$$

Moreover, we introduce the shared GRU network to recursively decode the frame-based correlation, the event-based correlation and the fused correlation into the corresponding optical flows, respectively. We further impose the cross-modal flow consistency loss for the final results as follows:

$$\mathcal{L}_{flow}^{consis} = || \sum_{n=0}^{N} \sum_{t=0}^{T} F_{t,n} - F^f||_1 + \sum_{t=0}^{T} ||F_t - F_t^{ev}||_1, \tag{9}$$

where $F^f$, $F^{ev}$, $F_{t,n}$ denote the optical flows from the frame, event and their fusion, respectively. Note that the fused optical flows need to be accumulated to be consistent with the optical flow estimated from the frame. In addition, we enforce the photometric loss (Yu et al., 2016) on frames:

$$\mathcal{L}_{flow}^{pho} = \sum \psi(I_t - I_{t+dt}(x + F^f)) + ||F^f - F^{gt}||_1, \tag{10}$$

where $\psi$ is a sparse $L_p$ norm ($p = 0.4$). This loss is also suitable for event-based optical flow. Therefore, motion correlation fusion module introduces an optimization solution to provide a self-supervised learning paradigm for the fusion network, explicitly modeling the frame-event spatiotemporal fusion process, thereby achieving robust dense and continuous optical flow.

Table 2: Quantitative results on the proposed dataset with daytime and nighttime conditions.

| Method | | ARFlow | Selflow | RAFT | GMA | E-RAFT | BFlow | **ComST-Flow** |
|---|---|---|---|---|---|---|---|---|
| Input | | Frame | Frame | Frame | Frame | Event | Frame-Event | **Frame-Event** |
| Daytime | EPE | 29.17 | 20.02 | 5.51 | 5.37 | 4.85 | 4.04 | **3.78** |
| | F1-all | 78.71% | 54.97% | 24.17% | 23.35% | 19.50% | 17.23% | **15.14%** |
| Nighttime | EPE | 35.13 | 24.18 | 7.28 | 6.81 | 5.14 | 4.53 | **3.95** |
| | F1-all | 87.05% | 62.66% | 32.19% | 30.25% | 23.24% | 20.37% | **16.42%** |

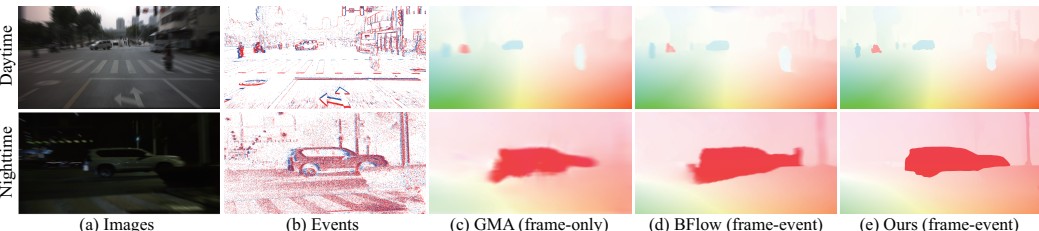

Figure 6: Visual comparison of optical flows on the proposed unseen dataset with various illumination.

## 3.4 Optimization and Implementation Details

Consequently, the total objective for the proposed framework is written as follows:

$$\mathcal{L} = \mathcal{L}_{flow}^{pho} + \lambda_1 \mathcal{L}_{kl} + \lambda_2 \mathcal{L}_{entropy} + \lambda_3 \mathcal{L}_{corr}^{spaErr} + \lambda_4 \mathcal{L}_{corr}^{tempErr} + \lambda_5 \mathcal{L}_{flow}^{consis}, \tag{11}$$

where $[\lambda_1, ..., \lambda_5]$ are the weights that control the importance of the related losses. The first term maintains the learning capability of frame-based and event-based optical flow estimators. The second and third terms locate visual boundary points as the reference template under the constraint of common spatiotemporal gradient. The fourth term makes the motion appearance spatially dense, and the fifth term guarantees the temporal continuity of motion trajectory. The final term constrains the optical flow consistency between different modalities. Regarding implementation details, we set the event slices $T$ as 20, the boundary class number $K$ as 10. During the training phase, we only need two steps. First, we use $\mathcal{L}_{flow}^{pho}$ to train feature encoders of frame and event modalities, and simultaneously use $\mathcal{L}_{kl}$, $\mathcal{L}_{entropy}$ to train visual boundary localization module. Second, we use $\mathcal{L}_{corr}^{spaErr}$, $\mathcal{L}_{corr}^{tempErr}$, $\mathcal{L}_{flow}^{consis}$ to train motion correlation fusion module for spatially dense and temporally continuous optical flow. During the testing phase, the final inference model only consists of frame-event encoders, feature pyramid network, transformer and GRU, thus achieving the end-to-end optical flow estimation.

## 4 Experiments

### 4.1 Experiment Setup

**Dataset.** We conduct extensive experiments on different datasets. Event-KITTI is a synthetic version of KITTI2015 dataset (Menze & Geiger, 2015), which generates the event stream corresponding to real images using v2e model (Hu et al., 2021). DSEC (Gehrig et al., 2021a) dataset covers different complex scenes, where we perform various degrees of blurry effect and frame extraction on images to simulate the spatiotemporal degradation, including Slow-DSEC and Fast-DSEC. In addition, we build a frame-event coaxial optical device, and adjust the exposure time and frame rate to collect the paired frame-event data with manually annotated optical flow labels under various illumination. Note that the proposed dataset is not used for training, but only for generalization comparison.

**Comparison Methods.** We divide comparison methods into unimodal and multimodal categories. Unimodal methods include frame-only (*e.g.*, unsupervised ARFlow (Liu et al., 2020) and Selflow (Liu et al., 2019b), supervised RAFT (Teed & Deng, 2020) and GMA (Jiang et al., 2021)) and event-only (*e.g.*, E-RAFT (Gehrig et al., 2021b)) approaches, while multimodal method (*e.g.*, BFlow (Gehrig et al., 2024)) requires frame and event data for fusion. For the comparison experiments, we have two training strategies for competing methods, one is to directly train the flow model on degradation images; the other is to first performs image restoration (*e.g.*, Deblur-GAN (Kupyn et al., 2018)), and then trains the competing methods on the restored results (named as "w/ Deblur"). Note that since the optical flow estimated by the proposed method is temporally continuous, it is necessary to accumulate optical flows with matched points along the time to obtain the final optical flow result for comparison.

Table 3: Ablation on main fusion losses.

| $\mathcal{L}_{kl}$ | $\mathcal{L}_{corr}^{spaErr}$ | $\mathcal{L}_{corr}^{tempErr}$ | $\mathcal{L}_{flow}^{consis}$ | EPE | F1-all |
|---|---|---|---|---|---|
| × | × | × | × | 1.21 | 5.09% |
| × | × | × | ✓ | 0.93 | 3.42% |
| ✓ | × | × | ✓ | 0.89 | 2.97% |
| ✓ | ✓ | × | ✓ | 0.78 | 2.56% |
| ✓ | × | ✓ | ✓ | 0.65 | 2.21% |
| ✓ | ✓ | ✓ | ✓ | **0.58** | **1.96%** |

Table 4: Ablation on backbones and fusion frameworks.

| Flow backbone | Fusion framework | EPE | F1-all |
|---|---|---|---|
| RAFT | w/o Fusion | 1.24 | 5.35% |
| | w/ Direct Fusion | 0.89 | 3.02% |
| | w/ Common Fusion | 0.60 | 2.08% |
| Flow Former | w/o Fusion | 1.21 | 5.09% |
| | w/ Direct Fusion | 0.84 | 2.74% |
| | w/ Common Fusion | **0.58** | **1.96%** |

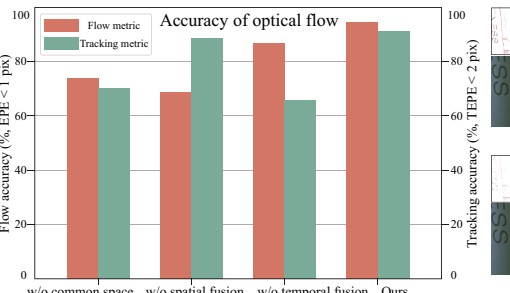
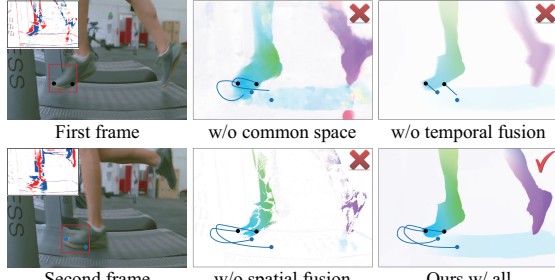

Figure 7: Effectiveness of common spatiotemporal fusion. Color and trajectory reflect the spatial and temporal performance of optical flow. Common space guides the cross-modal motion fusion, spatial fusion makes motion dense, and temporal fusion guarantees motion continuous.

As for quantitative evaluation, we choose average end-point error (EPE (Dosovitskiy et al., 2015)) and lowest percentage of erroneous pixels (F1-all (Menze & Geiger, 2015)) as the evaluation metrics.

## 4.2 COMPARISON EXPERIMENTS

**Comparison on Synthetic Datasets.** In Table 1, we compare competing methods with the two training strategies. First, the deblur strategy does indeed improve optical flow, but suffers the upper limitation. Second, multimodal methods outperform unimodal methods. This is because multimodal methods can fuse the complementary knowledge between various modalities. Compared with the other multimodal method, the proposed method with common fusion performs better.

**Comparison on Real Datasets.** In Table 1 and Fig. 5, we compare competing methods in real scenes with slow and fast motion patterns. First, there is no obvious difference between the performances of frame-based supervised methods and event-based methods in slow motion scenes. Second, event-based methods perform better than frame-based methods in fast motion scenes. Third, the proposed method with common fusion is superior to the multimodal method with direct fusion in real scenarios.

**Generalization for Unseen Dynamic Scenes.** In Table 2 and Fig. 6, we compare the generalization on the proposed dataset with various illumination. First, frame-based methods almost cannot work normally in nighttime scenes, while event-based methods still perform well. This is because event camera has the advantage of high dynamic range to model motion even in nighttime scenes. Second, compared with other event-based methods, the proposed method maintains significant generalization.

## 4.3 ABLATION STUDY

**How does Common Spatiotemporal Fusion Work?** In Fig. 7, we demonstrate the impact of common spatiotemporal gradient space, correlation spatial fusion and correlation temporal fusion on optical flow. Note that the y-axis labels denote the motion spatial accuracy (*i.e.*, flow EPE is less than 1 pix) and temporal accuracy (*i.e.*, tracking metric TEPE is less than 2 pix). Without common space, optical flow is spatiotemporally erroneous. Without spatial fusion, optical flow is spatially sparse but temporally continuous. Without temporal fusion, optical flow is spatially dense but temporally discontinuous. The common spatiotemporal fusion achieves dense and continuous optical flow.

**Effectiveness of Fusion Losses.** In Table 3, we conduct ablation study on the main fusion losses. $\mathcal{L}_{flow}^{consis}$ has a main positive effect on optical flow, but there exists an upper limit. $\mathcal{L}_{kl}$ slightly improves the flow results. $\mathcal{L}_{corr}^{spaErr}$ improves optical flow by fusing the spatially dense motion knowledge. $\mathcal{L}_{corr}^{tempErr}$ further improves optical flow by fusing the temporally continuous motion knowledge.

**Influence of Flow Backbone and Fusion Framework.** In Table 4, we discuss the impact of various backbones (*e.g.*, RAFT (Teed & Deng, 2020) and FlowFormer (Huang et al., 2022)) and fusion

Table 5: Discussion on clustering strategies.

| Strategy | EPE | F1-all |
|---|---|---|
| w/o clustering | 0.64 | 2.18% |
| w/ DBSCAN | 0.58 | 2.01% |
| w/ GMM | 0.59 | 1.98% |
| w/ K-Means | **0.58** | **1.96%** |

Table 6: Discussion on temporally tracking.

| Tracking target | EPE | F1-all | TEPE |
|---|---|---|---|
| w/o tracking | 0.74 | 2.43% | 2.89 |
| event signal | 0.71 | 2.35% | 2.74 |
| visual feature | 0.62 | 2.17% | 1.42 |
| correlation feature | **0.58** | **1.96%** | **1.14** |

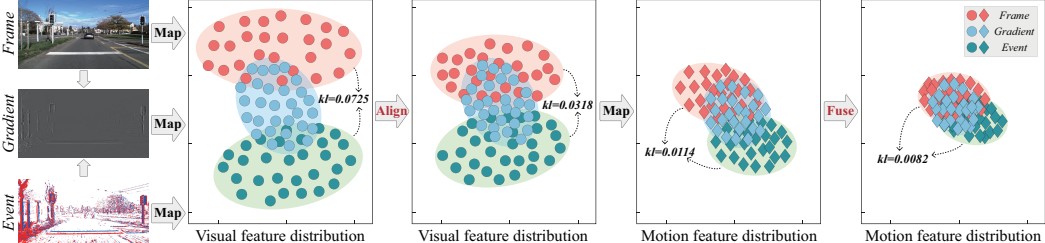

Figure 8: t-SNE visualization of visual and motion features. In visual feature space, common gradient can alleviate the feature discrepancy between frame and event modalities. In motion feature space, common gradient further guide the spatiotemporal feature fusion between the two modalities.

frameworks (*e.g.*, w/o fusion, w/ direct fusion and common fusion) on optical flow. For flow backbone, the network architecture has no obvious improvement on high-dynamic scene optical flow. For fusion framework, the spatiotemporal fusion is the key to estimate dense and continuous optical flow, and the common spatiotemporal fusion further improves the accuracy of motion feature fusion.

### 4.4 DISCUSSION

**Role of Common Space.** We analyze the role of common spatiotemporal gradient space in the feature alignment and fusion via feature visualization in Fig. 8. In visual feature distribution, frame-based and event-based features are mixed discrete, while common spatiotemporal gradient can bridge the two modalities. In motion feature distribution, common spatiotemporal gradient acts as an intermediate bridge, which further makes the motion features between the two modalities close.

**Choice of Clustering Strategies.** In Table 5, we discuss the impact of various clustering strategies (*e.g.*, DBSCAN (Ester et al., 1996), K-Means (Likas et al., 2003) and GMM (Reynolds et al., 2009)) on the spatial density of correlation feature. First, the fused optical flow without clustering is lower than those with clustering. Second, there is no difference between the impacts of various clustering strategies on the final optical flow. This reveals that correlation feature possesses the discriminative properties for motion patterns, which can be distinguished by simple clustering strategies.

**Why to Temporally Match Correlation.** We compare the effects of different tracking solutions (*e.g.*, event signal, visual feature and correlation feature) on the temporal continuity of optical flow in Table 6. First, tracking feature is significantly better than tracking event signal. This is because event signals are simple data with only positive and negative polarities, which is difficult to represent scene knowledge. Second, tracking correlation feature performs better than tracking visual feature. The reason is that correlation feature can directly reflect the temporal correspondence of optical flow.

**Limitation** The proposed method may suffer challenges for the radial moving object relative to the camera. This is because frame and event cameras can only capture x,y-axis motion patterns, while the radial moving object moves along the z-axis, making the two cameras ineffective for this special motion pattern. In the future, we will introduce LiDAR to assist in modeling the radial motion.

## 5 CONCLUSION

In this work, we propose a novel common spatiotemporal fusion framework between frame and event modalities to achieve dense and continuous optical flow. We construct a common spatiotemporal gradient space as an intermediate bridge to mitigate the heterogeneous data representation between frame and event modalities. The common space locates the visual boundary templates, which is used to explicitly guide the fusion of the frame-based spatially dense correlation and the event-based temporally continuous correlation. Moreover, we build a pixel-aligned frame-event dataset, and demonstrate that the proposed method significantly outperforms the state-of-the-art methods. We believe that our work could facilitate the development of high-dynamic scene optical flow.

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
