# OpenReview forum: "Bridge Frame and Event: Common Spatiotemporal Fusion for High-Dynamic Optical Flow"
_ICLR.cc/2025/Conference — ICLR 2025 Conference Withdrawn Submission_

### Official Review · Reviewer_qZKD · 2024-10-18

**Soundness:** 3
**Presentation:** 3
**Contribution:** 3
**Rating:** 5
**Confidence:** 4

**Summary:**

This paper proposed a frame-event fusion framework for high-dynamic optical flow estimation. A spatiotemporal fusion bridge between frame and event modalities is established via the gradient space. A pixel-aligned frame-event dataset with an optical coaxial device is constructed. Experiments on Event-KITTI, DSEC, and the established coaxial dataset demonstrate the effectiveness of the proposed method.

**Strengths:**

1. The paper is overall well-written and nicely structured.
2. Extensive ablation studies are conducted, which verify the design choices in the proposed framework.
3. The feature visualizations are interesting, as they show the guidance effects of the proposed intermediate bridge.
4. A coaxial frame-event dataset is established, which will help facilitate future research in frame-event optical flow estimation.

**Weaknesses:**

1. It would be nice if some visualization or numerical results could be presented to show the time-continuous optical flow estimation effects, not limited by the used dataset, for analysis. Some time-dense event-based optical flow estimation methods could be compared.

2. It would be nice if some frame-event fusion methods as discussed in the related work section, such as the cross-modal attention module and existing multimodal fusion methods, could be compared and analyzed in the experiments as well, which will better show the superiority of the proposed fusion solution.

3. If it is possible, a table can be added to compare the properties of existing event-based optical flow estimation datasets and your proposed one.

4. More time-dense-relevant results like inference time, latency, inference time of the proposed model could be discussed and analyzed.

**Questions:**

Would you consider discussing the recently appeared CoSEC coaxial stereo event camera dataset? How does your proposed dataset differ from existing datasets?

Would you consider presenting more computational costs of the proposed method and the components in the framework? This is relevant for real-world applications.

On L252, as event data indicates the log change of intensity, why is there a 'U'? Please clarify this.
On L264-269, how did you exactly calculate the similarity metric between the boundary maps with Euclidean distance? How did you obtain the histograms? These should be clarified.
On L289-294, how did you exactly generate the similarity distribution between the boundary features using convolutions? This should be clarified, as it is not detailed in Fig. 2.
On L360-365, do the two matchings indicate the results of optical flow warping? This should be explained.

---

### Official Review · Reviewer_Fw1n · 2024-10-29

**Soundness:** 2
**Presentation:** 1
**Contribution:** 2
**Rating:** 5
**Confidence:** 3

**Summary:**

The paper introduces a method for optical flow estimation in high-dynamic scenes using spatiotemporal fusion of event and RGB data. This method has two key components: visual boundary localization and motion correlation fusion to ensure dense and continuous flow estimation. Experiments on both synthetic and real datasets demonstrate that the proposed method outperforms compared approaches.

**Strengths:**

The motivation of the paper is well explained, and the method has better performance than the compared approaches.

**Weaknesses:**

- The method appears highly engineered; however, the language used to describe implementation details is vague. The paper lacks a detailed description of the method. The authors are encouraged to provide additional method details or introduce their approach mathematically in the main paper. For example, in Line 289, it is unclear how $p$ is obtained. Similarly, the reference to "$\mathbf{y}$ as the pre-classified boundary class label" could be clarified with more specific details. Additionally, any statistics on the collected dataset would be valuable.

- The compared state-of-the-art methods were re-trained (Line 428). However, in optical flow estimation, performance is highly dependent on the quantity of pre-training data. A model pre-trained on extensive datasets like FlyingThings, Sintel, KITTI-2015, and HD1K may achieve strong performance in high-dynamic scenes. If this is the case, the proposed method’s advantages may diminish. Additionally, the authors are encouraged to explore more recent unimodal methods beyond GMA, such as Flowformer, Flowformer++, and VideoFlow.

- In the DSEC benchmark, one test sequence is recorded at night. It would be beneficial for the method to be evaluated on the nighttime sequence within the DSEC benchmark to assess its performance.

**Questions:**

Please see the weaknesses.

---

### Official Review · Reviewer_VD8t · 2024-10-30

**Soundness:** 4
**Presentation:** 4
**Contribution:** 3
**Rating:** 8
**Confidence:** 4

**Summary:**

This paper focuses on optical flow estimation in high-dynamic scene, which suffers large displacement. The authors analyze the spatiotemporal degradations of frame-based optical flow under high-dynamic conditions, which motivates them to introduce the event camera with the advantage of high temporal resolution to assist in addressing this issue. Moreover, considering that existing methods have the risk of suffering the modality gap, the authors explore a common-latent space as an intermediate bridge to mitigate the modality gap, and propose a common spatiotemporal fusion framework for high-dynamic scene optical flow, including visual boundary localization and motion correlation fusion. The two modules can not only relieve the cross-modal feature discrepancy, but also make the fusion process interpretable to achieve dense and continuous optical flow. In addition, the authors conduct extensive experiments to verify the effectiveness of the proposed method.

**Strengths:**

1. The high-dynamic scene optical flow the authors focused on is a challenging problem.
2. The proposed common spatiotemporal fusion framework is significantly meaningful and promotes the spatiotemporal complementary fusion between frame and event cameras. This idea is novel to the optical flow community, providing a new paradigm for dense and continuous optical flow.
3. The motivation is logical. The authors analyze in detail the necessity of common space and spatiotemporal fusion, which improves the readability of the entire paper.
4. The figures are clear and nice.
5. The supplementary and demo promote the completeness of this work. Furthermore, the authors also provide a high-dynamic optical flow dataset. I hope that the authors can release this dataset.

**Weaknesses:**

1. The whole framework looks complex, including six losses. How to train the proposed method? I suggest that the authors can provide a detailed description.
2. Efficiency comparison of the proposed method, including runtime and model size.
3. The introduction about the proposed dataset is relatively short. The authors should add the details of this dataset.

**Questions:**

Please answer the questions in the weaknesses section.

---

### Official Review · Reviewer_47oC · 2024-11-02

**Soundness:** 2
**Presentation:** 2
**Contribution:** 3
**Rating:** 5
**Confidence:** 5

**Summary:**

This paper targets the task of optical flow estimation by combining images and events, by discussing similarity modeling in gradient space and spatio-temporal fusion in correlation construction of image and event features to estimate dense and continuous optical flows. Overall the framework is promising, but the paper lacks some necessary detailed explanations, and the experiments do not fully demonstrate the advantages of image degradation inputs as well as for continuous optical flow tracking.

**Strengths:**

1. As the modality difference between images and events, the authors construct boundary consistency from gradient to establish the association between them, which is used to construct the reference template.

2. The correlations constructed from events and images are fused to obtain enhanced time-continuous and spatially dense correlations, which leads to continuous dense optical flow estimation.

**Weaknesses:**

1. An established method [1] is closely related to the study of this paper, please present the differences with it.

2. $L_{flow}^{pho}$ includes both self-supervised and supervised parts, which maintains the learning capability of frame-based and event-based optical flow estimators. Whereas $F^f$ is the optical flow from frames, it is necessary to explain how Eq(10) can be used for event-based optical flow.

3. Event-based correlations are computed in Section 3.3 for estimating continuous optical flow, and implementation details need to be clarified. a) Are correlations computed from the first event slice to the subsequent, or adjacent slices? b) Are multi-frame continuous optical flows starting from frame 1, or between adjacent slices? c) How many continuous optical flows are estimated by the GRU and how are they accumulated to obtain the final optical flow result for comparison? Does the aggregation process need to deal with the occlusion of invisible points? d) The performance of temporally continuous optical flow is only represented by the TEPE metric in the ablation experiment section and is not compared with existing methods such as BFlow[2]. In addition, the differences between the image event correlation fusion of this paper and BFlow need to be clearly stated.

4. The overall training process uses K-means clustering and EKF tracking as well as image gradients and Canndy contours several times. It is necessary to clarify whether these are required to ensure gradient backpropagation in model training.

5. Slow-DSEC and Fast-DSEC are obtained by adding various degrees of blur effect to DSEC. Event-KITTI is a synthetic version of the KITTI 2015 dataset with simulated events, but significant blurring can be observed in Table 5 of Supp, which is inconsistent with the original KITTI dataset. The implementation details of adding blur effect need to be clarified, whether based on multi-frame blending or on motion aggregation. Both KITTI and DSEC have publicly available leaderboards. At least submission on a real-captured event dataset, DSEC, is necessary.

6. Lack of explanation of training details, including hyperparameters, training dataset, etc. Do the compared methods take the same training setup as the method in this paper? The compared image-based optical flow estimation methods are outdated. Since this paper introduces the transformer for correlation fusion, at least some transformer-based new methods such as GMFlow[3] should be compared.

[1] Exploring the Common Appearance-Boundary Adaptation for Nighttime Optical Flow, ICLR 2024
[2] Dense continuous-time optical flow from event cameras, TPAMI 2024
[3] GMFlow: Learning Optical Flow via Global Matching, CVPR 2022

**Questions:**

1. The real dataset collected in this paper has optical flow ground truth. The details of labeling the optical flow of dynamic objects need to be stated.  There are a few high-quality optical flow datasets available due to the difficulty of optical flow collection and labeling. Will the authors make this dataset publicly available?
2. Eq(4) builds on image gradients and contours, which leads to two questions: Do gradients and canny edges computed on degraded input images find edges accurately? Are L_kl and L_entropy built only between two frames and do not take full advantage of continuous events?
3. What F_t stands for in Eq(9)?

---

### Note · Authors · 2024-11-17

I have read and agree with the venue's withdrawal policy on behalf of myself and my co-authors.